# Transition from undergraduates to residents: A SWOT analysis of the expectations and concerns of Japanese medical graduates during the COVID-19 pandemic

**Mikio Hayashi**[1]*, **Katsumi Nishiya**[1], **Kazunari Kaneko**[2]

**1** Center for Medical Education, Kansai Medical University, Osaka, Japan, **2** Department of Pediatrics, Kansai Medical University, Osaka, Japan

* h_mikio106@hotmail.com

**Data Availability Statement:** In this study, qualitative data were used. Due to ethical restrictions related to protecting participants privacy, the full dataset cannot be shared publicly.

## Abstract

### Introduction

Interruptions in undergraduate clinical clerkship during the COVID-19 pandemic have reduced the confidence and preparedness of residents beginning their postgraduate training. We explore the thoughts of new residents about this transition and reflect on the support needed.

### Methods

An exploratory qualitative case study was conducted with 51 residents. All had experienced interruptions in clinical training due to the pandemic and had just started their postgraduate training. Qualitative data were collected through 6 focus groups and 12 individual follow-up interviews. A thematic analysis was undertaken, and the data were categorised using a Strengths, Weaknesses, Opportunities, and Threats (SWOT) framework.

### Results

Graduates beginning their residency were aware of their professionalism and independence during the transition. They also faced the predicament of needing close supervision while their supervisors managed pandemic conditions. Residents emphasised the importance of developing relationships with colleagues and supervisors during the transition to residency and wanted direct observation and detailed feedback from their supervisors during procedures.

### Conclusions

The experiences of residents were not uniformly negative. In fact, some had developed a positive mindset when entering the clinical field. Medical faculty members reflecting on interactions with new residents and planning future clinical internships could benefit from placing a high value on building relationships among residents, who may expect direct observation and detailed feedback from their supervisors.

However, the minimal dataset is available upon request to the Institutional Review Board of Kansai Medical University Hospital via kmuccr@hirakata.kmu.ac.jp for researchers who meet the criteria for access to confidential data.

**Funding:** The authors received no specific funding for this work.

**Competing interests:** The authors have declared that no competing interests exist.

## Introduction

'I had not seen patients for a long time because of the interruption in clinical clerkship because of the pandemic, so I was afraid of interacting with them at first when I went out into the clinical field. . . Although there is a risk of contracting COVID-19 in the clinical field, I hope to improve my skills so that I can help patients soon.'

The prolonged interruption of clinical training during undergraduate medical education, due to the COVID-19 pandemic, has profoundly impacted the clinical experience of medical students. Students have reported a reduction in their confidence and preparedness as they have begun their postgraduate clinical training [1, 2]. Previous studies on the transition to residency before the pandemic revealed that residents felt inadequately prepared for practice, perceived a lack of organisational support, and could only assess their readiness with reference to their own experience [3, 4]. Therefore, the transition to practice has been, for the most part, viewed negatively by residents, who have found interpersonal support difficult to access [5]. It was also found that early clinical experiences and shadowing opportunities seemed to have a positive impact on the transition, but inappropriate interactions with the supervisors had a negative impact on the readiness [4]. As medical students prepare themselves to meet the real demands of being a physician and face the challenges of transitioning to a new role as a resident, supervisors must provide the right kinds of learning and supportive environments to positively impact the interns during their transition [6]. Various improvements have been made over the years to make this transition as smooth as possible, including improved placement opportunities in the medical university, initiation of assistantships, and paid shadowing during induction [7]. Continuous mutual adaptation and preparation of the individual, as well as the preparation of the host environment and a strong support network, are necessary and can help maximise the opportunities that can be provided while minimising potential negative impacts [8]. During the pandemic, practical guidelines were published to support the transition from under- to post-graduate medical education, which not only addressed the need to tailor residency programmes according to local needs and resources but also recommended an approach for each transition to residency period [9]. However, a recent Best Evidence Medical Education (BEME) review shows that medical faculties must determine what works, for whom, and under what circumstances [10]. When reviewing resident competency acquisition under pandemic conditions, faculties can address trainee-specific issues and impacts by working directly with residents regarding the co-creation of new expectations and procedures [11]. In contrast, medical students' return to clinical practice is complicated by their sense of professional responsibility and perceptions of potential risks to the healthcare system, indicating the need to qualitatively elucidate the details and derive new perspectives for the future, under pandemic conditions [12].

Conventionally, residents felt most prepared for history-taking and performing physical examinations, and least prepared for providing safe prescriptions to and early management for patients with emergency conditions [4, 13]. Further, because of the perception that clinical leadership is hierarchical and due to a lack of support or feedback from their peers and seniors, residents did not understand the role of management and felt unprepared regarding non-clinical skills [14]. Therefore, it is important to develop a common understanding of preparedness for practice among residents and stakeholders [3, 15]. It is also evident that the support of fellows of a similar age and the effective use of simulation-based education can improve the readiness of the residents [16, 17]. With regard to the transition to residency during the COVID-19 pandemic, residents who graduated early, because of a sense of responsibility or financial

motivation and went on to work on the front lines, perceived their experience as positive and felt that they should be given future opportunities to monitor and improve their skills [18, 19]. Although there is little research on how the mismatch between residents and supervisors in the transition to residency can be addressed [20], we can bridge this gap by exploring the thoughts and perspectives of such residents—who experienced interruptions in undergraduate clinical training due to the pandemic and had just started their postgraduate training—to establish the necessary support. Carefully considering and addressing these issues will assist in the planning for residency under pandemic conditions. Our aim is not to generalise or predict but to create a deeper understanding of the phenomenon being investigated. The findings will serve as a guide for medical faculties when considering how to interact with residents and planning future clinical internships.

## Methods

### Study design

We followed the Standards for Reporting Qualitative Research recommendations [21] and used an exploratory qualitative case study methodology, positioned within a constructivist paradigm [22, 23]. The strength of this approach is the consideration of various perspectives of a phenomenon, allowing for in-depth investigation using multiple sources and methods for data collection [24, 25]. This approach provided us with detailed information to interpret the perspectives of the residents. In this study, we conducted focus group and individual follow-up interviews to identify the participants' perspectives on their transitional experiences during the COVID-19 pandemic. We used theoretical sampling [26] to understand the thoughts of participants whose undergraduate clinical clerkship was interrupted due to the COVID-19 pandemic. The study protocol was developed in accordance with the ethical guidelines at Kansai Medical University Hospital and within the Declaration of Helsinki. Ethical approval for this study was provided by the Institutional Review Board of Kansai Medical University Hospital (2020315).

### Context

All medical universities in Japan offer six-year programmes. Formal clinical clerkships are typically conducted in the fourth, fifth, and sixth years of study. Medical students are required to complete two years of postgraduate clinical training and pass the national examination. Graduates in Japan do not have a speciality; their speciality is selected after they complete two years of postgraduate clinical training. Japan, like many other countries, has faced the challenges of the pandemic since January 2020. The first state of emergency was declared in April 2020. Most educational institutions cancelled or suspended clinical clerkships, and some implemented remote education for medical students [27]. In Japan, the government recommended that universities should operate resiliently, based on general guidelines; the requirements of medicine for experiential and practical components, such as clinical clerkships and anatomy practice, were not addressed. Medical universities made decisions appropriate for their local contexts. At Kansai Medical University Hospital, where this study was conducted, the content of clinical clerkship was flexibly adjusted according to the local COVID-19 infection status. Alternative remote clinical clerkship was the main content of clinical training, and direct contact with patients was not allowed, even during clinical training in the hospital. In addition, clinical clerkship at the community hospital was cancelled, as were elective clerkships outside the hospital. In Japan, simulation training was conducted in some institutions, but because of the risk of infection in such training, most universities did not use simulators in the early stages of the COVID-19 pandemic and practised clinical clerkship remotely. However, at the

time of residency orientation, vaccination was available, and training with the simulator was permitted in most institutions. There were few specific restrictions on postgraduate clinical training, except in departments that treated COVID-19 patients.

## Participants

We recruited graduates in the first year of their postgraduate residency who had experienced interruptions in their undergraduate clinical training. All employed first-year residents in the Kansai Medical University Hospital were invited for voluntary participation. Overall, 51 graduates (mean age 26.5 years; range 24–41 years) agreed to participate. Of them, 12 also participated in individual follow-up interviews. The participant profiles are provided in Table 1. The authors established a good rapport with the participants through group work during resident orientation and simulation training prior to the residency training. All participants were informed of the aim of the focus group (i.e. to understand and explain the meanings, beliefs, and cultures that influence the participants' feelings and behaviours). We informed the participants that the interview data would be kept confidential and that they would be used only for the purpose of this study. For this, we procured written consent from all participants.

## Data collection

To understand the residents' thoughts and perspectives on the transition from under- to postgraduate medical training, the authors conducted 6 focus groups, consisting of 7–9 residents each, during orientation for residency training. Focus groups allow for the discussion of complex topics and emphasise the interactions between research participants to generate data and explore the participants' thought processes [28]. The focus groups were face-to-face and semi-structured—to clarify participants' perspectives on their transitional experiences—and conducted by the first author in April 2021. Key topics were: 'What are your expectations and concerns about postgraduate clinical training, having experienced the conditions of the COVID-19 pandemic in undergraduate medical education?' and 'Based on your situation, what kind of support do you think is necessary for the residents' transition, given their experience of interruptions in clinical training during undergraduate medical education, due to the COVID-19 pandemic?' In addition, 12 semi-structured follow-up interviews were conducted—within two months of the focus groups—to understand further details during their transition. The interview guide for each of the interviews is illustrated in Table 2. Questions used in the interviews were formulated based on previous literature [28]. The interview guide was then revised on the basis of the pilot testing carried out by the researchers (MH and KN), and several residents had volunteered to participate before the focus group interviews began. On the basis of the pilot test, the authors decided that question 2 in the focus group should be changed to 'expectations and concerns' rather than just 'thoughts' in order to elicit as much diversity as possible

**Table 1. Characteristics of study participants.**

| Characteristic | Focus group (%) | Individual interview (%) |
|---|---|---|
| **Gender** | | |
| Female | 25 (49) | 6 (50) |
| Male | 26 (51) | 6 (50) |
| **University** | | |
| Kansai Medical University | 31 (61) | 6 (50) |
| Other Medical University | 20 (39) | 6 (50) |
| **Total** | 51 | 12 |

**Table 2. Interview guide.**

| Focus Group |
|---|
| 1. How was your undergraduate medical university experience influenced by the COVID-19 pandemic? |
| 2. What are your expectations and concerns regarding postgraduate clinical training with regards to the COVID-19 pandemic? |
| 3. Based on your experience, what kind of support do you think is necessary for clinical residents? |
| Semi-structured follow up interviews |
| 1. In which departments are you currently being trained? |
| 2. How have you been spending your time since the start of your clinical training? |
| 3. What are your concerns and expectations with regard to the training? Why? |
| 4. What kind of support do you need through your training experience in clinical settings? And why? |
| 5. Were there any transition-related issues that arose in the daily interactions among residents after the focus group interview? |

from the residents. The interview guide was not revised after the focus group interviews started, because the authors agreed that it was suitable to be used for the research purpose. The focus groups and interviews were all conducted in Japanese and audio-recorded. Data were transcribed verbatim immediately after each interview and translated into English by the authors.

## Data analysis

The focus group and interview data were thematically analysed, including generative coding and theorising to identify instances of similar concepts in the dataset [29, 30]. An inductive coding approach was applied by MH and KN, in several iterations, until an agreement was achieved. In addition, the third author (KK) checked the results. The themes were categorised into main and subcategories and tabulated using NVivo11 (QSR International, Australia) to identify their frequency. In case of a disagreement, the authors discussed and reviewed the data until a consensus was reached. After the themes emerged, the authors applied a Strengths, Weaknesses, Opportunities, and Threats (SWOT) analysis to group them into four categories: internal strengths, internal weaknesses, external opportunities, and external threats. SWOT analysis is a tool that reflects an individual or organisation's current position and highlights opportunities for growth [31]. It has been applied in medical education to identify adaptations and strategies in response to COVID-19 [32, 33]. Our analysis revealed the latent and complex inner struggles experienced by residents whose clinical training had been interrupted.

## Results

Through the SWOT analysis of data collected from the focus groups and follow-up interviews, we identified themes regarding residents' expectations, concerns about training, and required support during COVID-19. The themes and sub-themes which emerged are summarised in S1 and S2 Tables, with some exemplary quotes.

## Strengths

**Positive mindset.** Residents who had experienced interruptions in clinical clerkship during undergraduate training because of the pandemic thought that the experience of living with restraint, behavioural restrictions, dealing with uncertain situations, and coping with stress strengthened their mental health and resilience during the transition to residency. They also believed that their awareness of how their behaviour might affect the patients of the affiliated

hospital, during undergraduate medical education, would help them develop their professional awareness.

**Awareness regarding self-management.**  Although residents who experienced the pandemic as medical students, felt anxious about going out into the field because of a lack of clinical experience, they realised that the practice of strict infection control measures and health management while in medical school had strengthened their personal and collective awareness of infectious diseases. They also believed that their personal health management during COVID-19 would contribute to self-reporting and prevention of burnout, in case of illness during their residency training.

## Weaknesses

**Concerns pertaining to clinical competency.**  Residents felt that they lacked the skills to establish good patient-doctor relationships, because of the limited opportunities to see patients in clinical settings during their undergraduate medical education. They realised that it was difficult to form a rapport when they could not see the other person's expression because of infection control measures (masks, PPE). They not only realised the differences between the simulation experiences in undergraduate medical education and the procedures they performed on actual patients, but also reaffirmed their concerns about the accuracy of their physical examination skills.

**Unfamiliarity with the clinical environment.**  The residents who had experienced interruptions in clinical clerkship felt a strong sense of unfamiliarity with their work when they went into the field, and thought that it would take significant effort to understand the routine work of the departments and resident duties during their rotations. They also felt that they had to be careful when talking to their supervisors, nurses, and other professionals who were busy with their daily work under pandemic conditions.

## Opportunities

**Room for growth.**  Residents felt that the interruption in their training had helped them develop an attitude of independent learning. However, they also felt that it is necessary to acquire methods to apply the medical knowledge learned during medical university to clinical practice through future training and practice. They appreciated the opportunity to learn substantially while taking care of each patient through careful questioning because they could feel that their clinical skills were developing through their interactions with actual patients. They also realised that the tension of the clinical setting during COVID-19 would help them develop as residents when they oversaw patient treatment.

**Cooperation with others.**  There was also a determination to build relationships with supervisors and an emphasis on community among residents. They believed that the exchange of information and encouragement among residents would lead to mutual success in residency training. Further, they felt that their experiences in the pressurised clinical setting of the pandemic, fostered in them a desire to learn quickly to become immediate assets in the field.

## Threats

**Inadequate socialisation.**  Residents were concerned that the interruption of electives and off-campus training might reduce their opportunities to experience cases and procedures. They were also concerned about the possible narrowing of career choices due to limited opportunities under pandemic conditions. Further, they viewed the conventional clinical clerkship as a preparation period for adapting to the clinical environment, and thought that the

interruption of clinical training due to the pandemic might negatively impact their behaviour as members of society.

**Uncertainty during the COVID-19 pandemic.**   Moreover, residents immediately experienced stress associated with strict infection control measures after entering the clinical field, and felt physically and mentally burdened by the demands. They were also concerned about the risk of not only contracting COVID-19 themselves but also infecting their patients and other residents by starting their training before they were vaccinated.

### Desired support for transition

**Specific instructional methods.**   Residents wanted an environment in which their supervisors provided step-by-step guidance, considering their lack of clinical experience. They also felt that it was important to experience procedures under the direct observation of their supervisors and receive detailed feedback on their clinical practice.

**Training support system.**   Residents who had just started their clinical rotations thought that it was important for them to be provided with a clear training policy and an extensive manual, to compensate for their lack of experience in each speciality. They believed that instructions from their supervisor on how to gather information efficiently and support from fellows of a similar age would help them develop as residents.

**Cordial relationship.**   Further, the residents expected to be approached by their supervisors and believed that a sense of closeness and candour was important in their relationship with them. They also believed that support without discrimination was desirable, regardless of gender, nationality, or school of graduation.

## Discussion

Our case study demonstrates that graduates who have just started their residency programme are aware of their professionalism and independence during the transition. They also face the predicament of needing close supervision while their supervisors are managing pandemic conditions. The findings of this study compensate for the lack of clarity about the complexity of professional responsibility and inner expectations in previous studies [12]. Each supervisor can utilise the results as a basis for discussion in their context, when planning the adjustments necessary for graduates starting their residency under pandemic conditions. Further, the details of expectations and concerns regarding the transition to residency amid the pandemic were revealed. Although SWOT analysis has been useful in analysing organisational management in previous studies under COVID-19 [33], we suggest adopting it for future planning of support for residents in each context.

The results of this case study will be useful for two reasons: first, it will help to support residents beginning their training; and second, it will be useful when planning adjustments to undergraduate medical education during the pandemic. Although the recent BEME review proposes measures to ensure undergraduate medical students continue to engage in safe, in-person clinical learning [1], the same considerations should be made for the transition to residency. Several themes contribute to residents' leadership, including expectations of community-building. Conventionally, it has been assumed that new residents are reluctant to play a leadership role, and have limited motivation for teamwork and collaboration [14]. Conversely, our findings suggest that the residents who experienced disruption due to the pandemic may enter the clinical field with a positive mindset. Faculty members should consider these characteristics when placing them in internships. Additionally, the potential strengths of residents, identified through this study, may inform the planning of residency beyond the pandemic period.

Previous studies on the transition to residency have shown that residents tend to feel prepared for history-taking and performing physical examinations [4, 13]. However, the results of this study suggest that residents who have experienced disruption in clinical clerkship, during their undergraduate medical education due to the pandemic, not only lack confidence in these areas, but also feel anxious about patient contact. Although graduates perceive that their readiness to work can be improved by increased independence after a few months in a new position [34], direct observation by their supervisors and detailed feedback should be implemented to alleviate their concerns. Although direct observation is recognised as a valid and reliable assessment tool, some learners feel ambivalent about being observed [35], and supervisors should consider this. We have shown that residents seek a sense of closeness with their supervisors, and a learner-centred approach to interpersonal and teamwork skills facilitates this [20]. Additionally, our results confirm that the support from fellows who are not much older than the residents is helpful [16] and should be maximised. A useful future investigation would be on the extent to which this approach reduces the sense of isolation and improves the preparedness that residents experience during transitions. The evolution of positive and negative aspects can also be traced.

From the results of this study, we believe that medical school faculty members can focus on the positive aspects of the COVID-19 pandemic to encourage self-improvement among medical students who are unable to gain sufficient clinical experience and provide more substantial learning opportunities, including online materials. In addition, medical school faculty may be able to motivate medical students by providing them with feedback on their clinical assignments from a detailed clinical perspective. The aforementioned efforts have the potential to facilitate the transition to residency. In addition, providing opportunities to strengthen relationships among classmates may lead to more fulfilling clinical preparation.

This study has some limitations. First, the results may not be transferable, because this study was conducted in one medical university hospital, with residents who had just started their residency programme. Medical universities in Japan follow a Model Core Curriculum, and we believe that there are no substantial differences in the curricula depending on the type of university. However, it is necessary to consider that many of the study participants were from private universities. Further research is required in other contexts, including nationwide or multi-country surveys, for validation. In addition, we believe that similar studies should be conducted not only in university hospitals but also in community hospitals. A second limitation is that it lacks a long-term perspective. Transition to residency should not be viewed as a moment in time, but as a future opportunity for professional development that requires support [5]. Continuous research regarding the challenges and adaptations in the transition to residency is required, as the pandemic continues to affect residents' experiences. Finally, although this study adopted a thematic analysis, congruent with the constructivist stance, the realist approach or professional identity formation framework may reveal other perspectives on residents' expectations and concerns about postgraduate clinical training, given the effect of the pandemic on undergraduate medical education.

## Conclusions

In this study, we used SWOT analysis to identify the expectations and concerns of new residents—whose undergraduate clinical training had been interrupted by the COVID-19 pandemic—regarding their transition to residency. The experiences of the residents were not uniformly negative. In fact, some developed a positive mindset when entering the clinical field. The residents whose residency programme had just started felt that they lacked the skills required for establishing good patient-doctor relationships, because of the limited

opportunities which were available to attend patients in clinical settings during their under-graduate medical education. They are also aware of their professionalism and independence during the transition. During their transition, residents wanted direct observation and detailed feedback on their clinical practice from supervisors, to compensate for their lack of clinical experience. They also emphasised the importance of forging relationships with colleagues and supervisors in the successful transition to residency. As new residents begin their training in future clinical settings, faculty members will need to not only adopt previous findings on the transition to residency, but also build relationships between residents and supervisors. As stated in the introduction, our goal is not to make generalisations or predictions, but to gain a deeper understanding of the phenomenon under investigation. Our findings can serve as a guide for the medical school faculty, while considering how to interact with residents and planning for future clinical practice. We hope that the results of this study will be further utilised to conduct a SWOT analysis of the transitions in their own contexts, because it will help to understand the expectations and concerns of the residents. A useful avenue for future studies will be to explore how the potential strengths and concerns of residents—identified through this study—are being utilised in the field, beyond the current pandemic.

## Supporting information

**S1 Table. Summary of themes and sub-themes from the SWOT analysis, with example quotes.**
(DOCX)

**S2 Table. Summary of themes and sub-themes depicting desired support for transition, with example quotations.**
(DOCX)

## Author Contributions

**Conceptualization:** Mikio Hayashi, Katsumi Nishiya.

**Data curation:** Mikio Hayashi.

**Formal analysis:** Mikio Hayashi.

**Investigation:** Mikio Hayashi, Katsumi Nishiya.

**Methodology:** Mikio Hayashi.

**Project administration:** Mikio Hayashi.

**Supervision:** Katsumi Nishiya, Kazunari Kaneko.

**Writing – original draft:** Mikio Hayashi.

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
