## [Decision Letter · Decision Letter 0]

7 Feb 2022

PONE-D-22-02532Transition from undergraduates to residents: a SWOT analysis of the expectations and concerns of Japanese medical graduates during the COVID-19 pandemicPLOS ONE

Dear Dr. Mikio Hayashi,

Thank you for submitting your manuscript to PLOS ONE. After careful consideration, we feel that it has merit but does not fully meet PLOS ONE’s publication criteria as it currently stands. Therefore, we invite you to submit a revised version of the manuscript that addresses the points raised during the review process.

This MS was reviewed by three specialists and few major concerns were ariased. Please revise the MS accordingly or listed as limitations.

We look forward to receiving your revised manuscript.

Kind regards,

Wen-Wei Sung, M.D., Ph.D.

Academic Editor

PLOS ONE

Reviewers' comments:

Reviewer's Responses to Questions

**Comments to the Author**

1. Is the manuscript technically sound, and do the data support the conclusions?

Reviewer #1: Yes

Reviewer #2: Yes

Reviewer #3: Partly

2. Has the statistical analysis been performed appropriately and rigorously? 

Reviewer #1: N/A

Reviewer #2: N/A

Reviewer #3: No

3. Have the authors made all data underlying the findings in their manuscript fully available?

Reviewer #1: No

Reviewer #2: No

Reviewer #3: No

4. Is the manuscript presented in an intelligible fashion and written in standard English?

Reviewer #1: Yes

Reviewer #2: Yes

Reviewer #3: Yes

5. Review Comments to the Author

Reviewer #1: The authors make clear that this is a qualitative study. Based on that criterion, the paper is sound enough, although a more quantitative analysis would be optimal. This leads to the issue related to this comment from the authors"

"The datasets generated and/or analysed during the current study are not publicly available, in order to protect the originality of our work so that we can continue evaluating future examinees. However, they are available from the corresponding author on reasonable request." I'll let the editor decide if that is acceptable. Other than that issue, the paper satisfactorily meets its objectives.

Reviewer #2: Thank you for the opportunity to review the paper.

Overall the paper is well written and easy to follow. The study was well designed, although the discussion section I think can be expanded.

I have several comments hopefully can further improve the paper:

1. "these findings" in the conclusions section in the abstract is a bit vague in terms of where it refers to. Perhaps make it more specific, what findings meant here.

2. page 5 line 83-86: perhaps needs to be moved towards the end of the introduction. I feel there is a bit of repetition between this sentence and the last paragraph in the introduction.

3. Why not maximum variation sampling used in the study? to include participants from various background to enable the attainment of comprehensive views of the topic being studied.

4. Some information in the study design section of the methods needs to be moved into participants and analysis sections?

5. How was the pilot study results and were there any changes made based on the pilot study.

6. Theme: room for growth  perhaps needs to be more specific in terms of what areas still have the room for growth.

7. Perhaps the implications of the study need to be further elaborated, for example in terms of the guidance for the medical school faculty, how can the findings serve as the guide, perhaps this is something to be elaborated in the discussion, before the limitations.

Reviewer #3: I have reviewed the article by Hayashi et al. on Japanese medical graduates and their transition to residency during the COVID-19 pandemic.

Several medical schools in the U.S. have simulation-based training, aiming to help smooth this transition. It is unclear if this is available in Japan, and if the 51 residents interviewed had this training. This needs to be included in the Context section.

The cohort belongs to Kansai Medical University Hospital in Osaka, a prestigious institution, nevertheless, assertions like “we explore the perspectives of residents who have just started their postgraduate training in Japan” refer to an extrapolation that cannot be made. Of these 51 residents, 61% came from Kansai, and 39% came from other universities. How different were in terms of their medical school curricula? How many were private? Some residents were older 41 y.o., so their vision may skew the findings…

Since the questions were broad, the focus group approach was the appropriate one.

Authors state that prior to the SWOT analysis “In case of a disagreement, the authors discussed and reviewed the data until a consensus was reached”. Dr. Hayashi is an Assistant Professor, while Dr. Nishiya (whose name is misspelled in the Title Page, should be Katsumi) is full professor. Having worked in Japan myself I wonder if a disagreement of two individuals at different ranks can be solved in a fair equitative way. Maybe a third party should have blindly solve any disagreement.

Since outcomes of the interviews were categorized, one would like to see at least a series of pi charts of these. And maybe cross analyzed with age, of coming from this or that university?

I could not find details on the specialty. Should I assume that these were all Pediatrics residents?

6. PLOS authors have the option to publish the peer review history of their article (what does this mean?). If published, this will include your full peer review and any attached files.

Reviewer #1: No

Reviewer #2: No

Reviewer #3: **Yes: **Jorge L. Cervantes

---

## [Author Response · Author response to Decision Letter 0]

1 Mar 2022

Reply for Editor:

The authors appreciate the editor’s comments. In the main document, the sentences highlighted yellow in the copy with tracked changes indicate segments that have been revised after incorporating the suggestions provided by the editor and the reviewers. In addition, we have modified the following information to the availability of data and materials as you pointed out.

‘In this study, qualitative data was used. Due to ethical restrictions related to protecting participants privacy, the full dataset cannot be shared publicly. However, the minimal dataset is available upon request to the Institutional Review Board of Kansai Medical University Hospital via kmuccr@hirakata.kmu.ac.jp for researchers who meet the criteria for access to confidential data.’

Reply for Reviewers

Reviewer 1:

Comment 1: The authors make clear that this is a qualitative study. Based on that criterion, the paper is sound enough, although a more quantitative analysis would be optimal. This leads to the issue related to this comment from the authors "The datasets generated and/or analysed during the current study are not publicly available, in order to protect the originality of our work so that we can continue evaluating future examinees. However, they are available from the corresponding author on reasonable request." I'll let the editor decide if that is acceptable. Other than that issue, the paper satisfactorily meets its objectives.

Response: The authors are grateful for the reviewers’ comments and valuable suggestions. In the revised version of the manuscript, the yellow-highlighted lines indicate segments that were altered after considering the reviewers’ input. As you pointed out, this is a qualitative study and no quantitative analysis was conducted. We are grateful for your evaluation of the paper as a qualitative study.

Reviewer 2:

Comment 1: Thank you for the opportunity to review the paper. Overall the paper is well written and easy to follow. The study was well designed, although the discussion section I think can be expanded. I have several comments hopefully can further improve the paper.

Response: Thank you for acknowledging that the manuscript has adhered to the scope of the journal. We appreciate the reviewer’s comments and suggestions. The sentences highlighted yellow in the final version of the manuscript indicate the segments that were altered to incorporate the reviewers’ suggestions.

Comment 2: "these findings" in the conclusions section in the abstract is a bit vague in terms of where it refers to. Perhaps make it more specific, what findings meant here.

Response: We have made the following corrections to ‘these findings’ in the conclusion of the abstract as you pointed out (Page 2, lines 39–42).

‘Medical faculty members reflecting on interactions with new residents and planning future clinical internships could benefit from placing a high value on building relationships among residents, who may expect direct observation and detailed feedback from their supervisors.’

Comment 3: page 5 line 83-86: perhaps needs to be moved towards the end of the introduction. I feel there is a bit of repetition between this sentence and the last paragraph in the introduction.

Response: The points you have noted have been included in the last paragraph of the introduction. We have also revised the text to avoid possible repetition (Page 6, lines 97–101).

Comment 4: Why not maximum variation sampling used in the study? to include participants from various background to enable the attainment of comprehensive views of the topic being studied.

Response: In this study, we mainly wanted to explore the thoughts of residents whose clinical clerkships were interrupted during their undergraduate medical education due to the COVID-19 pandemic. In consideration of those criteria, we felt it necessary to exclude residents who had not experienced a clinical clerkship interruption due to the COVID-19 pandemic during their undergraduate education or who had merely experienced a situation related to COVID-19. For the reasons mentioned above, we did not use maximum variation sampling in this study.

Comment 5: Some information in the study design section of the methods needs to be moved into participants and analysis sections?

Response: As you pointed out, the study design section under Methods contains detailed information on the participants and analysis methods; we have deleted the duplicate content and moved the relevant section to the Participants section under Methods (Page 8, lines 150–156).

Comment 6: How was the pilot study results and were there any changes made based on the pilot study.

Response: In the pilot study, we tried to elicit the honest thoughts of the residents, but concerns about postgraduate clinical training were mainly identified. Therefore, the authors decided that question 2 in the focus group should be changed to ‘expectations and concerns’ rather than just ‘thoughts’, to elicit as much diversity as possible from the residents. The above information has been added to the data collection section of the Methods (Page 9, lines 177–179).

Comment 7: Theme: room for growth  perhaps needs to be more specific in terms of what areas still have the room for growth.

Response: Residents believe that they need to pursue more direct communication with patients and self-directed learning style. This theme includes the meaning that it is necessary to acquire (through future training and practice) methods for applying the medical knowledge learned during medical university to clinical practice. It also means that residents could feel that their clinical skills were developing through their interactions with actual patients. We have added each of these descriptions to the theme of room for growth (Page 13, lines 240–246).

Comment 8: Perhaps the implications of the study need to be further elaborated, for example in terms of the guidance for the medical school faculty, how can the findings serve as the guide, perhaps this is something to be elaborated in the discussion, before the limitations.

Response: From the results of this study, we believe that medical school faculty members can focus on the positive aspects of the COVID-19 pandemic to encourage self-improvement among medical students who have been unable to gain sufficient clinical experience and provide more substantial learning opportunities, including online materials. In addition, medical school faculty may be able to motivate medical students by providing them with feedback on their clinical assignments from a detailed clinical perspective. The aforementioned efforts have the potential to facilitate their transition to residency. In addition, providing opportunities to strengthen relationships among classmates may lead to more fulfilling clinical preparation. The results of this study may serve as a link between medical school faculty and supervisors, and may lead to the development of complementary curricula that take transitions to residency into account. We have added the above information to the Discussion (Page 17, lines 327–337).

Reviewer 3:

Comment 1: I have reviewed the article by Hayashi et al. on Japanese medical graduates and their transition to residency during the COVID-19 pandemic.

Response: The authors are grateful for the reviewers’ comments and valuable suggestions. The yellow-highlighted lines in the revised version of the manuscript indicate the segments that were altered after considering the reviewers’ input.

Comment 2: Several medical schools in the U.S. have simulation-based training, aiming to help smooth this transition. It is unclear if this is available in Japan, and if the 51 residents interviewed had this training. This needs to be included in the Context section.

Response: In Japan, simulation training was conducted in some institutions, but because of the risk of infection in simulation training, most universities did not use simulators in the early stages of the COVID-19 pandemic and practised clinical clerkship remotely. However, at the time of residency orientation, vaccination was available and training with the simulator was permitted in most institutions. The above information has been added to the context in the Methods (Pages 7–8, lines 137–141).

Comment 3: The cohort belongs to Kansai Medical University Hospital in Osaka, a prestigious institution, nevertheless, assertions like “we explore the perspectives of residents who have just started their postgraduate training in Japan” refer to an extrapolation that cannot be made. Of these 51 residents, 61% came from Kansai, and 39% came from other universities. How different were in terms of their medical school curricula? How many were private? Some residents were older 41 y.o., so their vision may skew the findings…

Response: Medical universities in Japan follow a certain guideline called the Model Core Curriculum developed by the Ministry of Education, Culture, Sports, Science and Technology, and we believe that there are no substantial differences in the curricula depending on the type of university. However, as stated in the limitation section in the Discussion, this study was conducted at a private university hospital, and it is necessary to consider the fact that many study participants (38 out of 51) were from private universities. In addition, we believe that similar studies should be conducted not only in university hospitals but also in community hospitals. We have added some of these points to the limitation section in the Discussion (Page 17, line 340–346).

Comment 4: Since the questions were broad, the focus group approach was the appropriate one.

Response: Thank you for your positive feedback. The authors also believe that the focus group was appropriate for this study.

Comment 5: Authors state that prior to the SWOT analysis “In case of a disagreement, the authors discussed and reviewed the data until a consensus was reached”. Dr. Hayashi is an Assistant Professor, while Dr. Nishiya (whose name is misspelled in the Title Page, should be Katsumi) is full professor. Having worked in Japan myself I wonder if a disagreement of two individuals at different ranks can be solved in a fair equitative way. Maybe a third party should have blindly solve any disagreement.

Response: First of all, thank you for pointing out the misspelling; I have corrected the name of Dr Nishiya (Page 1, line 6). In addition, the results of this analysis were confirmed by the third author, Dr Kaneko, who has a different affiliation, and he confirmed that there were no differences in the results. As you pointed out, there may be a possibility of bias in the results between two authors with different ranks, but the all authors believe that the combined expertise of the two authors, in addition to the input of the third author, was suitable for this work. All authors frequently discuss any relevant issues with their colleagues in the same department on a regular basis and are devoted to standardizing their work as medical education professionals. The above information has been added to author contributions (Page 1, lines 17–18) and data analysis in the Methods section (Page 10, line 190).

Comment 6: Since outcomes of the interviews were categorized, one would like to see at least a series of pi charts of these. And maybe cross analyzed with age, of coming from this or that university?

Response: We thank you very much for your kind remarks; we discussed whether to incorporate a series of pie charts and cross-analyses. Ultimately, we decided not to add this type of content after we evaluated the length of the entire manuscript and considered its readability as a qualitative study. We also wanted to avoid creating confusion with quantitative research. The possibility of a mixed study was also discussed by the authors, but was abandoned in consideration of the overall volume and readability of the entire manuscript.

Comment 7: I could not find details on the specialty. Should I assume that these were all Pediatrics residents?

Response: Thank you for your important remarks. Residents in Japan do not have a speciality; their speciality is selected after they complete 2 years of postgraduate clinical training. Therefore, none of the study participants had a speciality; this is common in all hospitals in Japan. This fact has been added to the context in the Methods section (Page 7, line 123–125).

---

## [Decision Letter · Decision Letter 1]

18 Mar 2022

Transition from undergraduates to residents: a SWOT analysis of the expectations and concerns of Japanese medical graduates during the COVID-19 pandemic

PONE-D-22-02532R1

Dear Dr. Mikio Hayashi,

We’re pleased to inform you that your manuscript has been judged scientifically suitable for publication and will be formally accepted for publication once it meets all outstanding technical requirements.

Kind regards,

Wen-Wei Sung, M.D., Ph.D.

Academic Editor

PLOS ONE

Reviewers' comments:

Reviewer's Responses to Questions

**Comments to the Author**

1. If the authors have adequately addressed your comments raised in a previous round of review and you feel that this manuscript is now acceptable for publication, you may indicate that here to bypass the “Comments to the Author” section, enter your conflict of interest statement in the “Confidential to Editor” section, and submit your "Accept" recommendation.

Reviewer #2: All comments have been addressed

2. Is the manuscript technically sound, and do the data support the conclusions?

Reviewer #2: (No Response)

3. Has the statistical analysis been performed appropriately and rigorously? 

Reviewer #2: (No Response)

4. Have the authors made all data underlying the findings in their manuscript fully available?

Reviewer #2: (No Response)

5. Is the manuscript presented in an intelligible fashion and written in standard English?

Reviewer #2: (No Response)

6. Review Comments to the Author

Reviewer #2: Thank you for the revised version of the paper. I think all comments have been sufficiently addressed.

7. PLOS authors have the option to publish the peer review history of their article (what does this mean?). If published, this will include your full peer review and any attached files.

Reviewer #2: No

---

## [Editor Report · Acceptance letter]

22 Mar 2022

PONE-D-22-02532R1 

Transition from undergraduates to residents: a SWOT analysis of the expectations and concerns of Japanese medical graduates during the COVID-19 pandemic 

Dear Dr. Hayashi:

I'm pleased to inform you that your manuscript has been deemed suitable for publication in PLOS ONE. Congratulations! Your manuscript is now with our production department. 

Kind regards, 

on behalf of

Dr. Wen-Wei Sung 

Academic Editor

PLOS ONE